# Improving Nefiracetam Dissolution and Solubility Behavior Using a Cocrystallization Approach

**DOI:** 10.3390/pharmaceutics12070653

**Published:** 2020-07-09

**Authors:** Xavier Buol, Koen Robeyns, Camila Caro Garrido, Nikolay Tumanov, Laurent Collard, Johan Wouters, Tom Leyssens

**Affiliations:** 1Institute of Condensed Matter and Nanosciences, UCLouvain, 1 Place Louis Pasteur, B-1348 Louvain-la-Neuve, Belgium; koen.robeyns@uclouvain.be (K.R.); camila.carogarrido@student.uclouvain.be (C.C.G.); laurent.collard@uclouvain.be (L.C.); 2Namur Research Institute for Life Sciences (Narilis), Chemistry Department, UNamur, 61 rue de Bruxelles, B-5000 Namur, Belgium; nikolay.tumanov@unamur.be (N.T.); johan.wouters@unamur.be (J.W.)

**Keywords:** nefiracetam, solid state, cocrystals, solubility, dissolution rate, stability, formulation

## Abstract

In this work, we are the first to identify thirteen cocrystals of Nefiracetam, a poor water-soluble nootropic compound. Three of which were obtained with the biocompatible cocrystallization agents citric acid, oxalic acid, and zinc chloride. These latter have been fully structurally and physically characterized and the solubility, dissolution rate, and stability were compared to that of the initial Active Pharmaceutical Ingredient (API).

## 1. Introduction

Nefiracetam, a member of the racetam family, is a nootropic compound typically administered as a cognitive enhancer. This API ((*N*-(2,6-dimethylphenyl)-2-(2-oxopyrrolidin-1-yl)acetamide)) known under the label DM-9384 is a pyrrolidone derivative produced and developed during the 90s by Daiichi Seiyaku and marketed in 2002 in Japan [1,2]. Recently, a polymorph screen by our group [3] revealed the existence of three anhydrous polymorphs as well as a monohydrate form with the asymmetric units illustrated in Figure 1. Different solid-state forms exhibit distinct physicochemical properties such as the melting point, hygroscopicity, solubility, dissolution rate, and bioavailability. Polymorphs FI and FII are enantiotropically related anhydrous forms with FI found to be the most stable form below 140 °C, whereas FII is stable above this temperature. Polymorph FIII is metastable at all temperatures. Evaluation of the solubility/dissolution rate differences between polymorphs was challenging due to the fast solvent-mediated transformation of the meta-stable forms into the most stable one in suspension. 

Therefore, as polymorphism does not seem a reasonable approach to modulate the dissolution/solubility properties of this drug, due to the high-energy state of its polymorphs, we here investigate an alternative approach focusing on cocrystal and ionic cocrystal formation [4,5,6,7,8]. Such an approach seems promising for Nefiracetam as it is a non-ionizable, low-water soluble drug (Biopharmaceutical Classification System (BCS) II) displaying synthons favorable for hydrogen bonding and metal coordination, but rendering salt-formation impossible. The FDA guidance defines cocrystals as “solids that are crystalline materials composed of two or more different molecules generally held together by hydrogen bonds in the same crystal lattice” [9,10]. Considering that the molecular components have to be neutral and solid under ambient conditions, salts and solvates can be excluded from this terminology [9,11]. Ionic cocrystals (ICC) can also be included in this category with the specificity that they result from cocrystallization between a neutral organic molecule and an inorganic salt through complexation [12,13]. In the last two decades, cocrystallization has been widely used on pharmaceutical compounds to tune the solubility and/or dissolution rate [14,15,16,17], offering novel patentability opportunities in parallel [18]. Metaxalone cocrystals [19] with succinic acid, adipic acid, fumaric acid, salycilic acid, and maleic acid were patented in 2014 showing enhanced physicochemical properties. In 2006, Mc Amara et al. [20] showed that the use of glutaric acid as a cocrystallization agent increased the water solubility of a non-disclosed API about eighteen times and the dissolution rate about three times. Brittain et al. [21,22,23] recently reported the increasing importance of pharmaceutical cocrystals in the last decade. 

## 2. Materials and Methods

*Starting Materials.* Nefiracetam was purchased from Xiamen Top Health (Fujian, China). The solvents were sourced from VWR (Leuven, Belgium) and directly used without any purification steps. In the cases of ethyl acetate and acetonitrile, they were dried using a dessicant as the calcium hydride (CaH_2_). Nefiracetam was purified by a slurry crystallization in ethyl acetate (room temperature, ca. 300 rpm and overnight) and the solid phase was filtrated, washed, and dried. All coformers were commercially available from Alfa Aesar (Kandel, Germany), Acros Organics (Leuven, Belgium), TCI (Zwijndrecht, Belgium), and Merck (Overijse, Belgium) and used as received.

*Nefiracetam Cocrystals Screening.* A total of 133 cocrystallization agents typically used in cocrystallization efforts were selected among carboxylic acids, amides, sugars, inorganic salts and amino acids, other racetams, and profens. The cocrystal screening was performed through liquid assisted grinding (LAG) using a MM 400 Mixer Mill grinder manufactured by Retsch (Haan, Germany). The device is equipped with two grinding cells in which five 2 mL Eppendorf tubes can be set. To do so, equimolar amounts of Nefiracetam (0.2 mmol) and coformer (0.2 mmol) were weighted in an Eppendorf, and 4–5 stainless steel beads and 10 µL of solvent (methanol) were appended. Once the jars were filled, the milling runs for 90 min at 30 Hz.

For the three cocrystals studied in more detail in this work, these experiments were repeated using 12 different solvents in the LAG experiments as the nature of the solvent has been shown to impact on the outcome of such an experiment [24,25]. Using a 2:1 Nefiracetam/acid ratio for oxalic and citric acid and a 1:1 ratio for ZnCl_2_. The tested solvents were ethanol (EtOH), methanol, acetonitrile (MeCN), tetrahydrofuran, acetone, dichloromethane, chloroform, ethyl acetate, methyl acetate, diethylether, 2-propanol, and water, using approximately 10 µL of solvent.

*Single Crystal Growth.* Single crystals were mainly obtained from evaporative experiments. Nefiracetam and coformer were added in the molar ratio found in the cocrystal, and solids dissolved by the addition of a sufficient amount of solvent. Solutions were then left to evaporate slowly (over periods ranging from three to seven days) at room temperature, and single crystals retrieved. A 2:1 Nefiracetam-citric acid cocrystal Form I (NCA) and 1:1:1 Nefiracetam-zinc chloride-water ionic cocrystal hydrate (NZCW) suitable crystals were obtained from ethyl acetate, while the 2:1 Nefiracetam-oxalic acid cocrystal (NOA) crystals was from ethyl acetate and tetrahydrofuran. Cooling experiments were performed by preparing supersaturated Nefiracetam and coformer solutions. An excess of Nefiracetam and coformer (equimolar) was added to a given solvent volume at room temperature, and the vials were placed at 15 °C below the related solvent boiling point until the full dissolution. Once the full dissolution was achieved, they were stored at −15 or 9 °C in the case of water. Solid phases were then retrieved and analyzed. The 1:1 Nefiracetam-zinc chloride ionic cocrystal (NZC) suitable crystals were obtained from a cooling crystallization in dried acetonitrile.

*Nefiracetam Cocrystal Bulk Material Preparation.* The bulk material for dissolution testing was prepared through crystallization from the solution. To identify an appropriate solvent, slurrying experiments were performed by placing excess amounts of Nefiracetam-coformer in suspension in different solvents at 25 °C. Vials were sealed and the suspension was left over three days at 25 °C stirring at 700 rpm using a Cooling Thermomixer HLC manufactured by Ditabis (Pforzheim, Germany). Each vial was seeded with all possible solid forms (Nefiracetam and cocrystal) after 2 h of stirring. After three days, solid phases were retrieved and analyzed. The 12 aforementioned solvents were used for this section. A solvent was then selected in which the system behaves congruently (meaning a full transformation to cocrystal occurred over the three days), and an upscaled solvent-mediated cocrystal formation was performed. This upscaling was performed in an EasyMax 102 (Mettler Toledo, Zaventem, Belgium) crystallizer equipped with a hermetically closed 100 mL flask, under mechanic stirring (150 rpm) at 25 °C. After three days, the solids were retrieved, washed, and dried overnight at 50 °C. The solids were then used for dissolution measurements. NCA was upscaled to 30 g of the bulk material in ethyl acetate while NOA and NZC were upscaled in acetonitrile.

*Single-Crystal X-Ray diffraction (SCXRD).* Suitable crystals of NZCW to perform the SCXRD analysis have been analyzed using a Rigaku (Neu-Isenburg, Germany) Ultra X18S rotating anode, FOX3D mirrors. The diffracted beams were collected on a MAR345 image plate detector using MoKα (λ = 0.71073 Å). Single-crystal X-ray diffraction data for NCA, 2:1 Nefiracetam-citric acid cocrystal Form II (NCA1), and NZC (100 K) were collected on an Oxford Diffraction Gemini R Ultra diffractometer (Ruby CCD detector using CuKα radiation) and crystals of NOA were measured at the SNBL beamline (Pilatus 2 M hybrid pixel detector), ESRF Grenoble. All the methodologies for the data reduction [26], resolution and refinement [27], and validation [28] were specified as in our previous work [3]. The images of the crystal structures were drawn using the software Mercury 4.1.3 [29]. CCDC 2010261-2010276 contain the supplementary crystallographic data for this paper. These data can be obtained free of charge from The Cambridge Crystallographic Data Centre via www.ccdc.cam.ac.uk/structures.

*Powder X-Ray Diffraction (XRPD)* data were collected on a diffractometer Bragg-Brentano manufactured by PANalytical (Eindhoven, Netherlands). The X-Ray source, a Ni-filtered CuKα (λ = 1.54179 Å), was used at 40 kV and 30 mA. The detection was carried out using a X’Celerator detector. All the powders were analyzed in a 2θ angle range from 4 to 40° for a total scan time of 6 min 42 s (step size = ca. 0.0167°).

*Differential Scanning Calorimetry (DSC)* measurements were performed from 25 to 175 °C at a scanning rate of 5 °C·min^−1^ on a TA instrument DSC2500 (Zellik, Belgium) in the cases of NCA and NOA. The temperature range was extended to 275 °C for NZC and NZCW. Solid samples (6–7 mg) were placed in an aluminum crucible (40 µL) with pierced sealed lids and nitrogen was used as purge gas with a flow rate of 50 mL·min^−1^. Indium was used as a reference. Regarding the DSC measurements performed on the suspected and other confirmed cocrystals, the temperature range was extended to 200 and 225 °C.

*Thermogravimetric Analysis (TGA).* These analyses were carried out on a TGA-STDA 851e manufactured by Mettler Toledo (Columbus, OH, USA). The samples were analyzed in a temperature range from 25 to 400 °C (600 °C for NZC and NZCW). The scanning rate applied during the analysis was 10 °C·min^−1^. About 7–10 mg of solid materials were placed in an aluminum oxide crucible and a purge gas (nitrogen) was used with a flow rate of 50 mL·min^−1^.

*Dynamic Vapor Sorption (DVS)* analyses were performed at 25 °C on a Q5000 SA from TA instruments (New Castle, DE, USA). Solid samples (weight from 7 to 14 mg) were placed in a hanging platinum crucible and an empty crucible is used as the reference. These crucibles were placed in a chamber with controlled humidity and temperature. Nitrogen was used as flowing gas. Samples were first dried at 40 °C for 1 h before being exposed to variable relative humidity within a range of 10% to 90%. The equilibrium was considered to have been reached when the weight change was less than 0.010 mg·min^−1^ or no weight change has occurred for 2 h.

*Moisture Exposure.* Samples (100 mg) of NCA, NOA, and NZC were stored for one month at room temperature in a sealed box containing an open water flask. The atmosphere was assumed to be saturated in water (100% RH). Both the XRPD and DSC analyses were performed on those powders after the exposure.

*High Performance Liquid Chromatography (HPLC).* The calibration line (linear equation) for the Nefiracetam dosage was drawn by interrelating the area under the curve (AUC) with the concentration prepared. To do so, Nefiracetam samples were diluted 1000 times using a 1:1 acetonitrile-Milli-Q water diluent (in volume). Nefiracetam chromatograms were then recorded according to the HPLC parameters: Device: Waters Alliance 2695 (Zellik, Belgium); Column: Waters Sunfire C18, 4.6 × 100 mm, 3.5 μm; Detector: PDA 2998 (extraction at λ = 210 nm); T° = 40 °C; injection volume: 10 μL; flow: 1.2 mL/min; mobile phase A: H_2_O + 0.1% H_3_PO_4_; mobile phase B: CH_3_CN + 0.1% H_3_PO_4_; gradient: 0 to 0.5 min at 30% B; 0.5 to 4.5 min 30% B→90% B; 4.5 to 6.5 min at 90% B. The calibration line is reported in the Appendix A. 

*UltraViolet Spectroscopy (UVS).* The calibration line for UV spectroscopy was drawn in order to dose the Nefiracetam by matching the absorbance at λ_max_ (= 263 nm) and the concentration prepared with a linear equation. Acetonitrile was used as a diluent and blank. Sample UV-absorption spectra were recorded from 300 to 200 nm using a UV-1700 PharmaSpec spectrophotometer manufactured by SHIMADZU (Wemmel, Belgium).

*Dissolution Experiments.* Nefiracetam FI, NCA, NOA, and NZC were first ground using a mortar and pestle to get approximately the same particle size range. An excess amount of compound was added to 50 mL of solvent at 18 °C in a 100 mL flask under magnetic 100 rpm stirring. The 10 µL sampling occurred using a syringe equipped with a micro filter (0.02 µm). Sampling was performed every 10–15 s during the first minute up to t = 1 min, every 30 s up to t = 5 min, and every minute up to t = 15 min. A final sampling (triplicate) was performed after 24 h considering the equilibrium has been reached. Samples were diluted from 2000 to 5000 times depending on the fraction considered with a MeCN-water Milli-Q (1:1 in volume) diluent. Those fractions were then injected in HPLC and dosed as mentioned above. Concerning NZC, the 10 µL-fractions were diluted 250 times with MeCN before being dosed using a UV-1700 PharmaSpec (SHIMADZU) spectrophotometer. HPLC data as well as the UVS data related to the dissolution experiments are presented in the Appendix A.

## 3. Results and Discussion

### 3.1. Cocrystal Screening

A large cocrystal screen has been performed on Nefiracetam using liquid-assisted grinding (LAG) and 133 different coformers. A full list is shown in the Appendix A. Coformers have been selected based on crystal engineering principles, with all presenting synthons prone to cocrystal formation [30,31]. Seventeen cocrystals have been suspected based on XRPD data only, corresponding respectively to a success rate of 13%, which is in alignment with cocrystal success rates as reported in the literature [32]. Here, we report only those cases for which single crystals were obtained. This was the case for the following 13 coformers: 5-hydroxyisophthalic acid, 5-nitroisophthalic acid, 5-cyano-1,3-benzenedicarboxylic acid, 5-bromoisophthalic acid, 2-benzoylbenzoic acid, parahydroxybenzoic acid, (RS)-phenylsuccinic acid, (RS)-2-phenylbutyric acid, (RS)-3-phenyllactic acid, gallic acid, citric acid, oxalic acid, and zinc chloride (Figure 2). Cocrystals are also suspected by XRPD and DSC for the positional coformers 2,4-dihydroxybenzoic acid (β-resorcylic acid), 2,5-hydroxybenzoic acid (gentisic acid) and 3,4-hydroxybenzoic acid (protocatechic acid), and dipiconilic acid but no single crystal confirmation has been obtained so far.

In a more general manner, successful coformers usually tend to have at least one carboxylic acid group and a phenyl ring present. A similar trend has been observed for the cocrystallization of other racetam compounds [33,34]. Recently, we have also shown inorganic salts to be good cocrystal formers for racetam compounds [35,36], which is also confirmed here. Table 1 summarizes the main results for the 13 coformers leading to confirmed cocrystals. When the ground pattern does not match the simulated pattern from the single crystal, cocrystal polymorphs, solvates, or stoichiometrically diverse cocrystals are suspected [37,38]. This table also summarizes melting/dehydration temperatures of the Nefiracetam forms as well as the obtained cocrystals. Interestingly, we show that a very wide range of properties may be obtained using cocrystallization as a tool. Melting points vary from 68 to 243 °C merely by playing on the nature of the coformer. This shows the potential for variability one can achieve using cocrystallization as a crystal engineering tool. Furthermore, Table 1 also highlights the importance of screening for the various solid-state forms of a given API/coformer system, as cocrystal polymorphs or cocrystal hydrates can be readily encountered.

The cocrystals of citric acid, oxalic acid, and zinc chloride are described in more detail as these cocrystals are pharmaceutically acceptable (Figure 3). For the analysis of the others, we refer to the Appendix A. For citric acid (NCA), oxalic acid (NOA), and zinc chloride (***NZC***), LAG experiments all yielded the same outcome, irrespective of the solvent used. For these systems, a more profound form screening was performed using evaporative, slurrying, and cooling crystallization when applicable. Twelve solvents were used for a total of 125 experiments with results shown in the Appendix A. These results led to no additional crystalline form for the NOA cocrystal but did lead to the identification of a second 2:1 cocrystal form for the NCA system and a cocrystal hydrate for the NZC system, highlighting the importance of screening for multiple cocrystal forms once a positive coformer has been identified.

### 3.2. The 2:1 Nefiracetam-Oxalic Acid Cocrystal (NOA)

As mentioned above, only one form of the NOA cocrystal has been identified, which is easily identifiable through its 2θ peaks at 11.3, 13.6, and 15.9°. The bulk product pattern is furthermore shown to match the one simulated from the powder resolution analysis (Figure 4a), which shows NOA to crystallize in the monoclinic *P*2_1_ space group. Figure 5a shows a main hydrogen bond $C_{1}^{1}\;(4)$ chain pattern according to Etter’s graph-set notation [39,40] built through N-H···O (2.86 (1) Å, 152.0° as a hydrogen bond distance N···O and NHO angle) hydrogen bonds between Nefiracetam amide moieties. Nefiracetam is linked to the oxalic acid through a hydrogen bonding pattern between the hydroxyl group from both the carboxylic acid moieties and the oxygen of the γ-lactam moiety of two different Nefiracetam molecules. These interactions (2.54 (1) Å, 167 and 171.5°) result in a $D_{1}^{1}$ (2) hydrogen-bond pattern. This assembly creates an overall pattern with Nefiracetam chains along the *a*-axis and oxalic acid acting as a linker between chains (zig-zag feature in Figure 5a represented by a full black line). The main crystallographic parameters related to each form and the refinement parameters are summarized in Appendix A. Upon heating, NOA shows a sharp melting endotherm at 162 °C (Figure 6c) and right upon melting, a weight loss of about 15% is observed (Figure 6a), corresponding to oxalic acid sublimation [41,42]. After 200 °C, degradation of Nefiracetam is observed. After a prolonged (one-month) exposure under a humidity saturated atmosphere (100% RH), no transformation of NOA into another form or deliquescence were observed by XRPD and DSC measurements (Appendix A).

### 3.3. The 2:1 Nefiracetam-Citric Acid Cocrystal (NCA and NCA1)

Two different polymorphs of the 2:1 Nefiracetam-citric acid cocrystal have been obtained. All screening experiments led to the same 2:1 cocrystal form (NCA). Single crystals of this form were obtained from a slow evaporation from ethyl acetate. Unexpectedly, the SCXRD diffraction experiment of NCA revealed also the presence of a second 2:1 form NCA1, in the same crystal. This latter is likely due to a phenomena of cross-nucleation, i.e., the heterogeneous nucleation of Form II (NCA1, daughter form) on the surface of the other (NCA, parent form) [43,44]. This Form II was never observed in sufficient quantity in any of the screening or bulk material preparation experiments, and is therefore likely a kinetic form, crystallizing under high supersaturated conditions, which occurs at the end of the solvent evaporation process. As during upscaling, the crystallization solvent is never fully evaporated, and the material is washed, this polymorph is very likely not present in the bulk material that fits the NCA simulated powder pattern in Figure 4b. NCA crystallizes in the triclinic *P*-1 space group. As for the anhydrous forms of Nefiracetam [3], the Nefiracetam molecules are stacked in a chain, by intermolecular amide-amide hydrogen bonds (N-H···O, 2.773 (5) Å, 168.2°) leading to a $C_{1}^{1}\;(4)$ hydrogen-bond pattern. In Figure 5b, the recognition between Nefiracetam and citric acid is due to two extra $D_{1}^{1}\;(2)$ intermolecular hydrogen bonds (O-H···O, 3.18 (1) Å, 146.64° and O-H···O, 2.596 (5) Å, 168.01°) which link one citric acid molecule to two Nefiracetam molecules through the oxygen of the γ-lactam moiety (Nefiracetam) and the hydroxyl of a carboxylic acid (citric acid). Moreover, a ring feature $(R_{2}^{2}\;(8)$ with 2.654 (6) Å and 152.45°) leading to a cyclic citric acid dimer is observable between two carboxylic acids from different citric acid molecules. Finally, an intramolecular hydrogen bond O-H···O ($S_{1}^{1}\;$(3), 3.176 (7) Å, 121.48°) is present in each citric acid molecule.

Polymorph NCA1 shows disorder and crystallizes in the monoclinic *P*2_1_/*c* space group. Intermolecular interactions between citric acid and Nefiracetam are identical to those mentioned for the first NCA form (Figure 5c). In Figure 6a, TGA data shows three respective weight losses in the case of NCA. A first weight loss occurs around 80 °C, and must likely correspond to the loss of water molecules trapped in the channel-like structure observed for this cocrystal, evidencing that NCA is likely a non-stoichiometric hydrate. The following waves correspond to citric acid degradation (±28%, 175 °C) [45] and Nefiracetam degradation (from 225 °C), respectively. The DSC endotherm at around 80 °C corresponds to a potential dehydration of a non-stoichiometric hydrate, but could potentially also correspond to a melting point. DVS shows a deliquescent (+20% in weight) behavior at RH > 80%. When exposing NCA to a saturated humidity atmosphere for a long time period (one month), deliquescence is followed by Nefiracetam monohydrate crystallization (Appendix A).

### 3.4. The 1:1 Nefiracetam-Zinc Chloride Ionic Cocrystal (**NZC**)

The form screening with zinc chloride led to two different cocrystals. The full screening results are presented in the Appendix A. Slurrying crystallization in a congruent and dried solvent, led to a 1:1 ionic cocrystal (NZC) wherein Nefiracetam is coordinated with the Zn^2+^. In addition, evaporative experiments at room temperature with water traces in the solvent (ethyl acetate) led to a 1:1:1 hydrated ionic cocrystal (so called NZCW) form whose structure was solved showing a zinc chloride complex with Nefiracetam and one water molecule. Only NZC was upscaled in dried acetonitrile. We were unsuccessful at obtaining larger amounts of NZCW as all experimental conditions tried led to mixtures of the Nefiracetam monohydrate, NZCW, and anhydrous NZC. The experimental and simulated XRPD data related to each system are presented in Figure 4c. Single crystals obtained from slowly cooling a supersaturated acetonitrile solution, yielded suitable monoclinic *C*2/*c* cube-like single crystals of the anhydrate (NZC) wherein Nefiracetam is complexed to zinc chloride showing a tetragonal geometry around the Zn^2+^ ion. Nefiracetam molecules are bound to each other by two identical (by symmetry), tetrahedral complexes as shown in Figure 2 and Figure 5d (right). Complexation occurs between zinc chloride and the γ-lactam C=O moiety of a first molecule and the amide C=O from a second molecule and vice versa. The donor moiety (N-H) from the amide is involved in a hydrogen bond pattern ${(C}_{1}^{1}$(4), 3.496(1) Å, 146.49°) with the chloride from the tetragonal complex along the *a*-axis. These tetrahedral complexes form a network of Nefiracetam dimer molecules as shown with black diamonds in Figure 5d. The π-stacking interactions are present between the Nefiracetam aromatic moieties with a centroid-centroid distance of about 3.54°. This stacking may explain the higher density (1.637 Mg/m^3^) comparing NZC with NZCW.

NZCW crystallizes in the monoclinic *P*2_1_ space group. The amide-amide hydrogen bond between Nefiracetam molecules as described above remains present but does not lead to a $C_{1}^{1}\;(4)$ chain pattern as in the previous cases. Indeed, the Nefiracetam chains are interrupted by water molecules and Nefiracetam amide moieties are now involved in three $D_{1}^{1}\;(2)$ hydrogen-bond patterns with respectively a water hydrogen (O-H···H, 2.734 (2) Å, 174.33°), a second Nefiracetam amide moiety (N-H···O, 3.019 (3) Å, 164.80°), and with a chloride ion (N-H···Cl, 3.331 (2) Å, 169.63°). An infinite chain (O-H···Cl, $C_{1}^{1}\;(4)$, 3.154 (2) Å, 169.19°) is also observed involving the second hydrogen from the water molecule and a chloride ion. In the presence of zinc chloride, the carbonyl not belonging to the pyrrolidone group is coordinated to the Zn^2+^ cation as well as to the water oxygen atom leading to a tetrahedral complex. This complex is highlighted in Figure 5e and forms “wave” chains stacking Nefiracetam densely (1.514 Mg/m^3^) along the *b*-axis. NZC shows a single melting endotherm at 243 °C. On the other hand, NZCW shows a continuous dehydration behavior (coordinated-water loss of ±4.5% corresponds to 1 equivalent of water) up until 175 °C. The DSC analysis then shows an endothermic peak at 200 °C which potentially corresponds to the melting of another polymorphic form of the ionic cocrystal or is a mere effect of the water still being present in the DSC capsule. To observe the potential transformation of NZC into NZCW, NZC was stored for one month in a saturated humidity chamber at room temperature. Even though, the DSC analysis indicates traces of the hydrated form (small endotherm around 210 °C), XRPD shows no such transformation. If NZCW is present in the bulk, it would only be in trace amount (Appendix A). Therefore, NZC could be a potentially interesting form to market.

### 3.5. Solubility and Dissolution Profile of Nefiracetam Cocrystals

Ethanol and acetonitrile were selected to study the dissolution profile at 18 °C (100 rpm) of NCA, NOA, and NZC in comparison to the API (Nefiracetam FI). As the cocrystals considered here behave incongruently in water, with Nefiracetam monohydrate crystallizing out rapidly, we were unable to determine the apparent cocrystal solubility in this solvent. Therefore, we decided to perform the dissolution studies in EtOH and MeCN as we aim at underlining the potential impact cocrystals can have on physicochemical parameters. NCA, NOA, and NZC are incongruent in EtOH leading to a final slurry of Nefiracetam, whereas a congruent dissolution is observed in MeCN. In the case of an incongruent system in EtOH, a “spring-parachute” [46,47] behavior (Figure 7a) is expected meaning that the concentration reaches a maximum (“apparent solubility” [48,49]) before dropping to the solubility of the drug form crystallizing out (here Nefiracetam FI). The final solubility of Nefiracetam might, however, still be different as solution interactions will occur. In the present case, the cocrystal dissolution kinetics combined to the crystallization kinetics of Nefiracetam FI occur too rapidly for the “spring-parachute” behavior to be observable. Nevertheless, one notices a clear impact of the coformer on the solubility of Nefiracetam (Table 2) with the overall solubility at 18 °C observed in both cases around 750 mM (+40%) instead of 519 mM (Figure 7c) in EtOH. The presence of ZnCl_2_ on the other hand, reduces solubility to about 199 mM.

Table 3 also shows a true cocrystal solubility in MeCN where all cocrystals behave congruently. Here, one also clearly sees the importance of the coformer nature on the overall solubility, with citric acid slightly increasing the amount of Nefiracetam dissolved, whereas oxalic acid reduces this amount by about 30% in comparison to Nefiracetam FI. A striking drop in Nefiracetam present in the solution of about 90% is observed using the NZC cocrystal.

The changes in solubility also reflect in the dissolution rate. However, they are all less important in comparison with the dissolution rate of the parent compound. It should nevertheless be noticed that all solid forms dissolve very rapidly, with half of the maximum solubility reached in the first minute and full solubility reached in the first 5 min of adding powder to the reactor. Therefore, we expect the bioavailability to be strongly impacted by the type of solid form used, not for the impact on the time required to reach solubility, but rather by the impact on the solubility value.

## 4. Conclusions

Nefiracetam cocrystal screening led to the identification of 13 novel cocrystal systems. Among these, three biocompatible systems were identified (NCA, NOA, and NZC) and their physicochemical properties (stability, solubility, and dissolution) were evaluated and compared to those of the parent compound Nefiracetam. The zinc chloride ionic cocrystal showed an impressive improvement in thermal stability, but strongly reduced the solubility by about 90% in organic solvents in comparison with the parent drug. The use of dicarboxylic acid coformers can induce a complete different behavior, with citric acid improving the solubility, while oxalic acid reduces this latter when the cocrystal behaves congruently. For non-congruent systems, the presence of the coformer always increases the overall solubility. All cocrystals, as well as the parent compound, are rapidly dissolving. Overall, this study confirms that cocrystallization can be used as an effective tool to impact the physico-chemical properties of a drug compound. A relevant choice in the coformer can either help improve formulation stability, or bioavailability of the drug. The coformer selection is of utter importance, as strong variations can occur depending on the nature of the selected coformer.

## Figures and Tables

**Figure 1 pharmaceutics-12-00653-f001:**
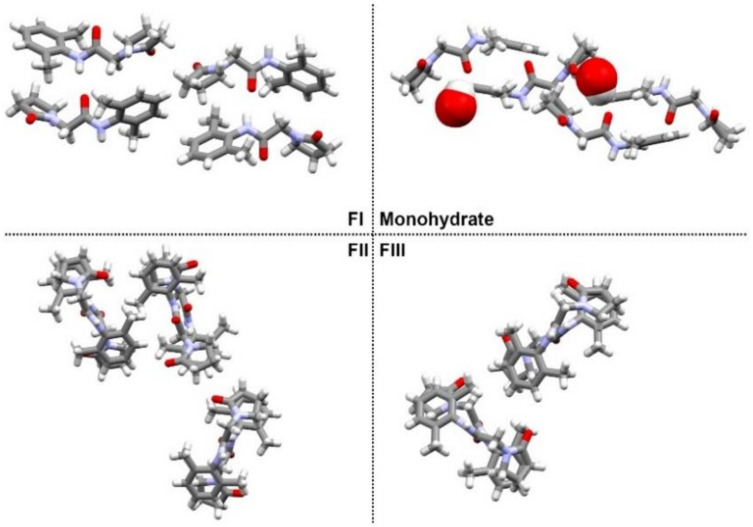
View of the crystal packing along the a-axis of each (pseudo)-polymorphic form of Nefiracetam (*N*-(2,6-dimethylphenyl)-2-(2-oxopyrrolidin-1-yl)acetamide). The spacefill representation is used to highlight water molecules in the monohydrate crystal structure.

**Figure 2 pharmaceutics-12-00653-f002:**
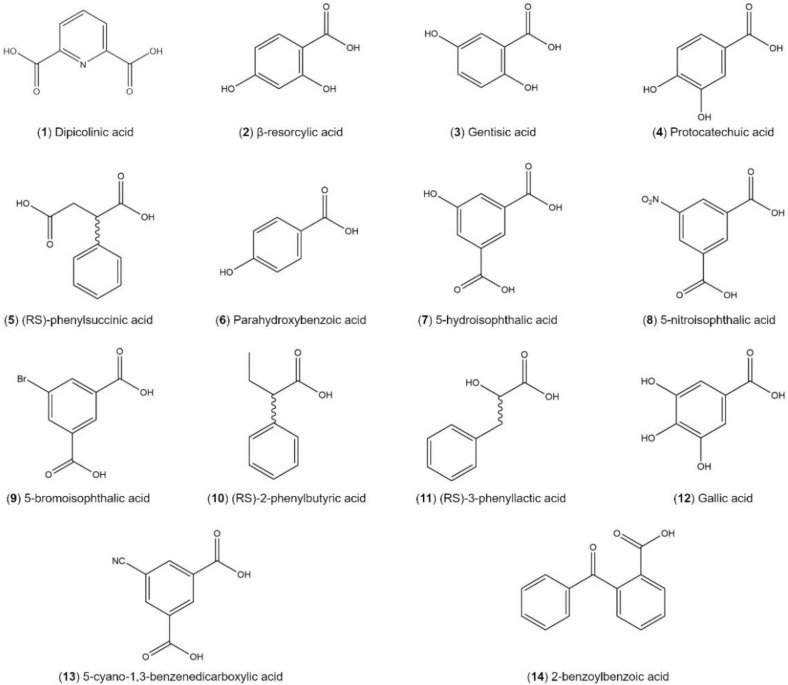
(**1**–**4**) Coformers for which a new Nefiracetam cocrystal is suspected and (**7**–**14**) coformers for which a single crystal cocrystal confirmation has been achieved.

**Figure 3 pharmaceutics-12-00653-f003:**
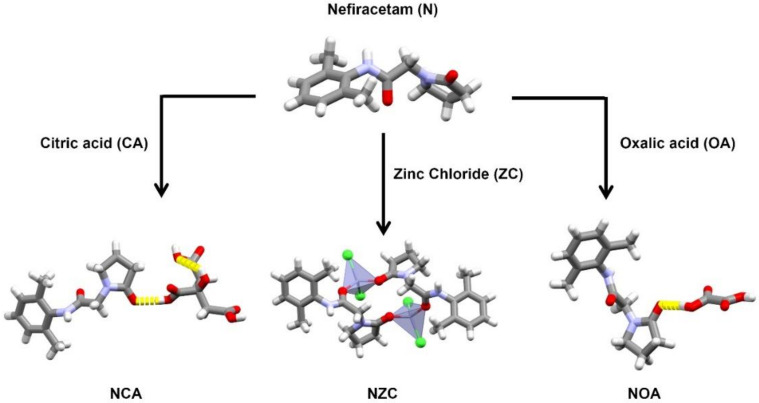
Biocompatible Nefiracetam cocrystals identified and studied in detail in this work. Relevant intramolecular interactions between Nefiracetam and the coformers are highlighted by yellow stick contacts.

**Figure 4 pharmaceutics-12-00653-f004:**
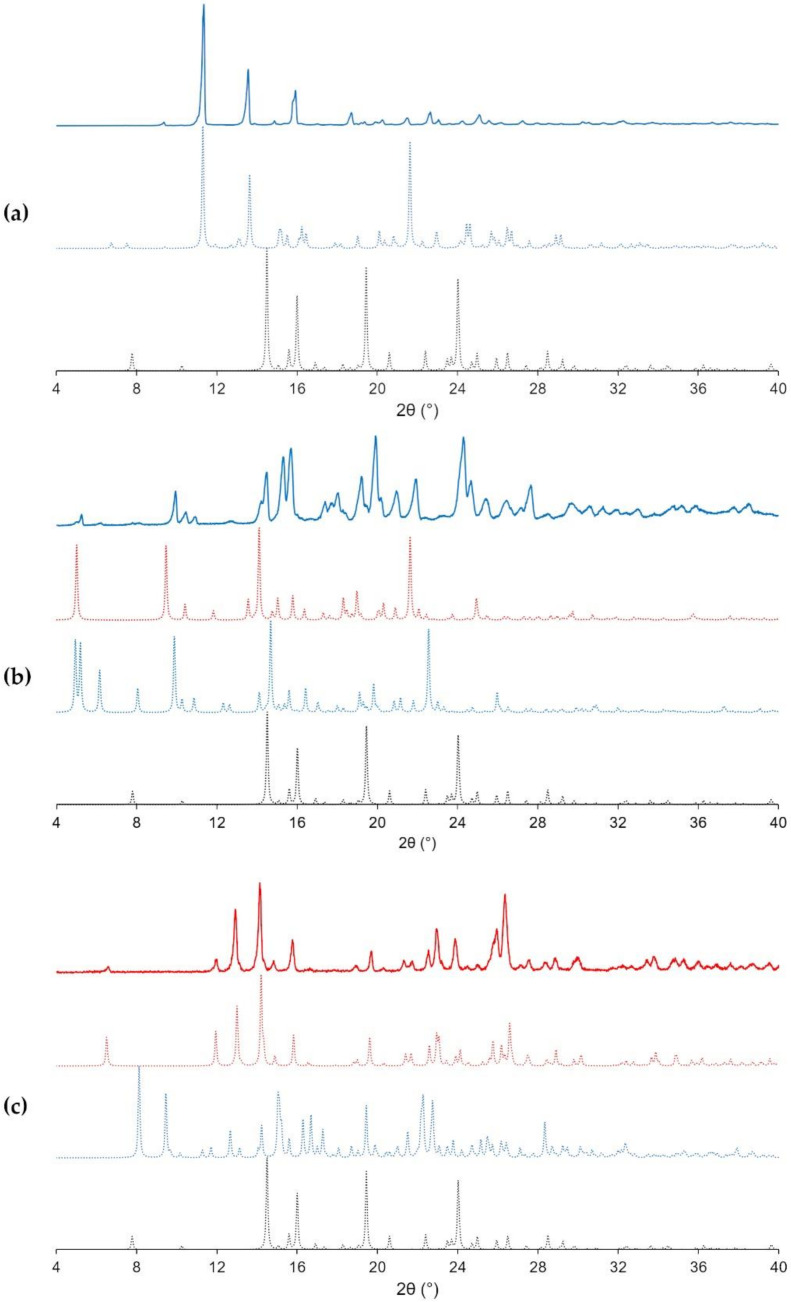
Simulated diffraction patterns (dashed line) and experimental powders (full line) of (**a**) Nefiracetam-oxalic acid cocrystal (NOA) (blue), (**b**) Nefiracetam-citric acid cocrystal (NCA) (blue) and NCA1 (red), and (**c**) Nefiracetam-zinc chloride ionic cocrystal (NZCW) (blue) and NZC (red). Patterns are compared to the Nefiracetam one (black).

**Figure 5 pharmaceutics-12-00653-f005:**
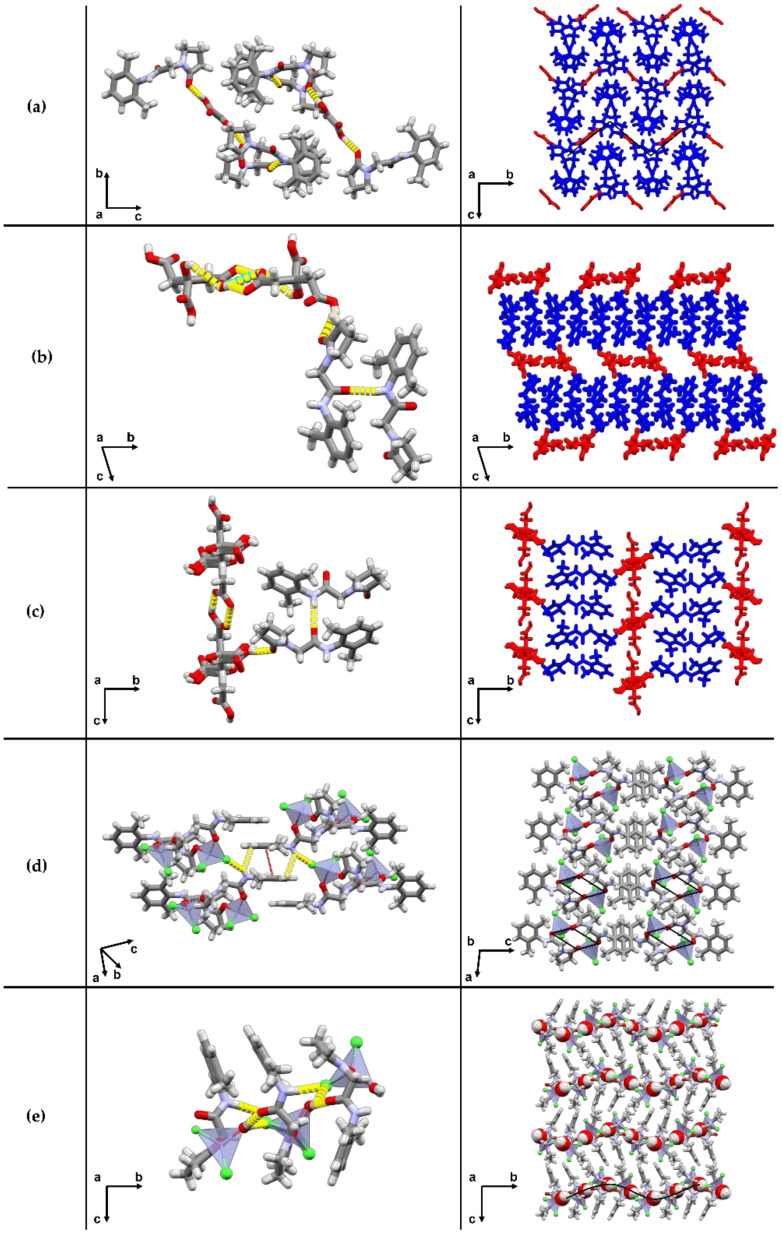
Main intermolecular interactions and crystal packing along the a- and b-axis in the crystal structure of NCA (**a**), NCA1 (**b**), NOA (**c**), NZC (**d**), and NZCW (**e**). The tetragonal geometry of the Zn^2+^ based-complex and the water molecules are respectively highlighted using the polyhedral and spacefill representation. Intermolecular contacts are shown using yellow dashed lines. The red dashed line is used to show the centroid-centroid distance in the NZC crystal structure.

**Figure 6 pharmaceutics-12-00653-f006:**
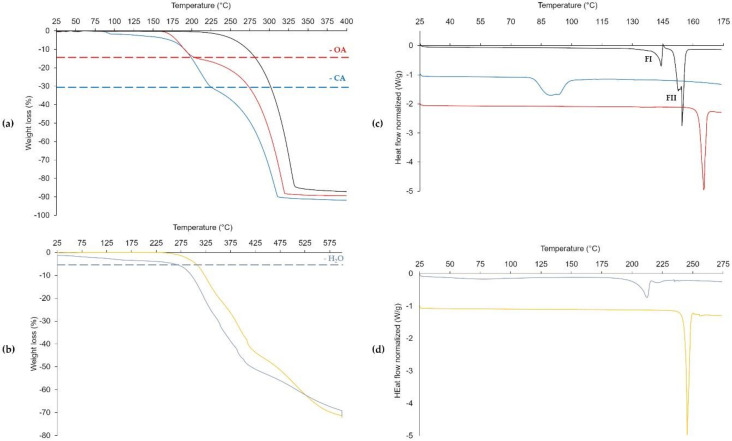
TGA curves (**a**,**b**) and DSC curves (**c**,**d**) of Nefiracetam (black), NCA (blue), NOA (red), NZC (yellow), and NZCW (grey).

**Figure 7 pharmaceutics-12-00653-f007:**
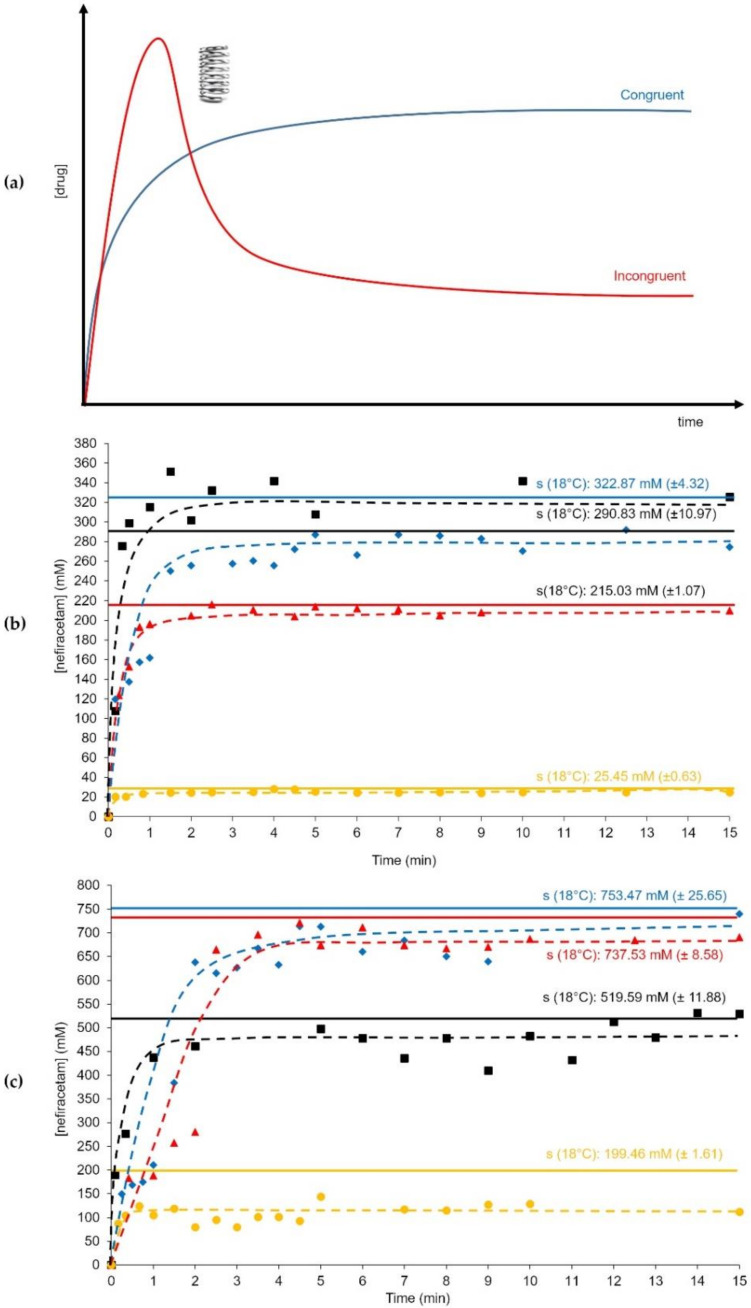
(**a**) Theoretical dissolution curve in the case of a congruent and an incongruent system (with spring-parachute behavior). Experimental dissolution curves of Nefiracetam FI (black ■), NCA (blue ♦), NOA (red ▲), and NZC (yellow ●) in (**b**) MeCN at 18 °C with a 100-rpm stirring and (**c**) EtOH at 18 °C with a 100-rpm stirring.

**Table 1 pharmaceutics-12-00653-t001:** Cocrystal forms identified, together with the melting point. Pharmaceutical systems of interest are highlighted in bold. RT: Room temperature; HT: High temperature.

Coformer	Multiple Cocrystal Forms	Form	Cocrystal/Coformer Melting Point (°C)
None [3] (Nefiracetam (N)/Figure 1)	/	Form I	140
		Form II	150
		Form III	n/a
		Monohydrate	80 (dehydration)
5-nitroisophthalic acid	Yes	1:1 Cocrystal Form I	171/259
		1:1 Cocrystal Form II	199
5-hydroxyisopthalic acid	No	1:1 Cocrystal	196/298
5-cyano-1,3-benzenedicarboxylic acid	Yes	1:1 Cocrystal Form I (RT)	160/248
		1:1 Cocrystal Form II (HT)	180
5-bromoisophthalic acid	No	1:1 Cocrystal	199/270
(RS)-phenylsuccinic acid	No	2:1 Cocrystal (racemic)	95/166
4-hydroxybenzoic acid	No	1:1 Cocrystal	122/213
(RS)-2-phenylbutyric acid	No	Form I (solid solution)	72/39
(RS)-3-phenyllactic acid	No	1:1 Cocrystal (racemic)	68/95
2-benzoylbenzoic acid	No	1:1 Cocrystal	92/126
Gallic acid	No	4:1:1 Cocrystal hydrate	n/a/251
Citric acid (CA)	Yes	2:1 Cocrystal Form I	81 (phase transition)
		2:1 Cocrystal Form II	n/a
Oxalic acid (OA)	No	2:1 Cocrystal	162
Zinc chloride (ZC)	Yes	Ionic cocrystal hydrate	210 (dehydration)/290
		Ionic cocrystal	243

**Table 2 pharmaceutics-12-00653-t002:** Dissolution rates and solubility at 18 °C in EtOH in the case of Nefiracetam FI, NCA, NOA, and NZC.

Solid Forms	Dissolution Rate * (mM/s)/Time (min)	Solubility (mM, 18 °C)
Nefiracetam FI	1299/0.2	519.59 (±11.88)
**NCA**	419/0.9	753.47 (±25.65)
**NOA**	246/1.5	737.53 (±8.58)
**NZC**	332/0.3	199.46 (±1.61)

* Dissolution rates correspond to the rate calculated when half the solubility is reached, with the corresponding time also given.

**Table 3 pharmaceutics-12-00653-t003:** Dissolution rates and solubility at 18 °C in MeCN in the case of Nefiracetam FI, NCA, NOA, and NZC.

Solid Forms	Dissolution Rate * (mM/s)/Time (min)	Solubility (mM, 18 °C)	Solubility Product (Ks, 18 °C)	pKs
Nefiracetam FI	969/0.1	290.83 (±10.97)	/	/
**NCA**	323/0.5	322.87 (±4.32)	1.35 × 10^−1^	0.87
**NOA**	538/0.2	215.03 (±1.03)	3.98 × 10^−2^	1.40
**NZC**	64/0.2	25.45 (±0.63)	6.48 × 10^−4^	3.19

* Dissolution rates correspond to the rate calculated when half the solubility is reached, with the corresponding time also given.

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
