# Peer review of "Improving Nefiracetam Dissolution and Solubility Behavior Using a Cocrystallization Approach"

_pharmaceutics, 2020, doi:10.3390/pharmaceutics12070653_

Round 1

Reviewer 1 Report

The authors did a great job on concisely presenting very interesting results of new multi-component solid forms of the API Nefiracetam. This work includes an extensive screening of coformers and further characterization of the new forms obtained. Scientific conclusions are supported and described properly. In my opinion, the work is competently carried out and could be attractive to Pharmaceutics readers. The work is worthy and may be considered for publication in its current state. A minor issue related to the reference software should be addressed in the manuscript.

Author Response

Dear Reviewer,

I'm very grateful to you for your very good feedback. I'm glad you appreciate the reading.

I'm aware there are some issues with my references and I will figure it out and change it for the final version. Thanks to have noticed it.

Thank you very much again

Have a nice day and stay safe

Xavier Buol

Reviewer 2 Report

The manuscript titled 'Improving Nefiracetam Dissolution and Solubility Behavior using a Cocrystallization Approach' by Tom Leyssens et al, is one of the best papers I have come across in recent days. This manuscript reported cocrystal, cocrystal polymorph, and ionic cocrystal with promising physicochemical properties. All the results are adequately represented, I couldn't find any flaws in this manuscript apart from very minor corrections. I hope this will become one of the hot papers in the Pharmaceutics journal.

Please correct the references as most of the places it is showing the error 'Error! Reference source not found' for example page 5/line number 194. Also, I would suggest giving some extra detail for figure 4 would be good as some of the powder patterns seems to mismatch with the simulated ones.

Also, I would imagine that the hydrate NZC ionic cocrystal may have formed by absorbing water from the atmosphere (during the crystallization) as you mentioned the solvents (including acetonitrile were dried on CaH2) in the materials and methods section or it could be solvent too. 

In the supporting information correct the spelling 'structures' for table 3, table 4, and table 5.

Author Response

Dear Reviewer

I'm uplifted by all the things you said about it. I'm glad you appreciate the reading and I thank you for the reviewing.

Issues about the Figures appear and I did not notice in time. I will correct it.

  • About the Figure 4 and the mismatching.

The NOA bulk powder (Figure 4a) has high preferential orientation (even  after grinding) and therefore the identification is clear with the triplet (11.3°, 13.6° and 15,9°). During the screening, this such XRPD pattern was always observed with some peaks "reduced". 

About the NCA bulk powder (Figure 4b), peaks  are broader probably due to the particles size ( I produce my powder in high  supersaturation conditions) and then the recognition is more difficult indeed. However, the main characteristic peaks below 15° in 2 theta fit with the NCA simulated one much more than the NCA1.

I hope I clarified a bit the situation for you. Do you wish I add one sentence or two in the manuscript about it ?

With NZC, there is competition in solution in presence of water between the nefiracetam monohydrate, the anhydrous NZC and the NZCW (hydrated NZC). In pure water, Nefiracetam monohydrate is definitely the one crystallizing out. With mixture of main acetonitrile with water, I got mixture of NZC and NZCW. Stoichiometric addition of water led to mixture of NZC/NZCW. While exposing NZC for one month in a 100% RH atmosphere, only the DSC reveals few % of NZCW but not enough to be observable in XRPD. NZC is quite resistant Under saturated humidity conditions then. Dried Acetonitrile was to avoid any traces of NZCW and to work with pure material.

Is it clearer for you ? Do not hesitate to contact me about it.

Thank you to have noticed my spelling mistakes in the Supporting Information. I correct it right now.

Thank you again for your reviewing and good words.

Have a nice day and stay safe

Xavier Buol

Reviewer 3 Report

In this work, the authors have conducted a comprehensive cocrystal screening of nefiracetam using more than 100 coformers. The screening resulted in 12 confirmed cocrystals and 5 novel solids yet to be identified as cocrystals or alternative forms. The novel solids have been characterized by X-ray diffraction, thermal and DVS analysis. Overall, the results are interesting and worthy of publication in ‘Pharmaceutics’, however, some section of the manuscript need further improvement. Therefore, I recommend revision of the manuscript as described below.

  1. Table 1. include melting point data of the coformers so that the impact of cocrystallization on the melting point can be easily understood.

  1. NCA1 does not represent a true polymorph of the cocrystal, 2:1 Nefiracetam-Citric Acid Cocrystal. TGA confirms that it is a hydrate and, therefore, this should be classified as a cocrystal hydrate and discussion associated with this should be revised accordingly.

  1. While it is convincing that the authors have used Ethanol and MeCN solvents as a medium for comparison of solubility and dissolution rate of the solid forms, but this data do not represent performance of these cocrystals in physiological conditions. Therefore, it is highly recommended that the authors should measure solubility and dissolution rate of the cocrystals in buffer solution and compare with the parent active. Apparent solubility/dissolution rate is a good measure of quantifying performance of dissociable multi-component solid forms. Hence, authors may follow the similar approach.

  1. The choice of Zinc chloride for making an ionic cocrystal should be justified.

  1. References are cited properly are missing in many parts of the manuscript and should be appropriately looked into it.

Author Response

Dear Reviewer,

First of all, I would like to thank you for your reviewing and the relevant comments you provide us. We appreciate your concern.

About your comments:

  1. Table 1: Coformers melting point

To add the reference mp of the parents is a great idea. It has been done like you suggested

Line 220

  1. NCA1: hydrate or polymorph?

NCA and NCA1 contain the same stoichiometry as evidenced from single crystal data. We were not able to obtain enough NCA1 to do a TGA. Indeed, NCA is likely a non-stoichiometric hydrate.

We added this to the text.

Line 292:

“…, evidencing that NCA is likely a non-stoichiometric hydrate”

This indeed does not allow us to really call NCA and NCA 1 polymorphs, so we adapted our table and the text and now call them. 2:1 Form I, 2:1 Form 2, removing the annotation to the word polymorph and just keeping the fact that they are different forms.

other corrections:

Line 230: “… a second 2:1 cocrystal form…”

Line 111: “…2:1 Nefiracetam-citric acid cocrystal Form II…”

Line 270: “a second 2:1 form NCA1…”

Line 271-273: “…the Form II (NCA1, daughter form) on the surface of the other (NCA, parent form). “This Form II…..”

Line 290: “…the first NCA form”

  1. Dissolution in Physiological conditions

You are perfectly right, and indeed these dissolution studies are ideally performed in physiological conditions. However, the cocrystals studied here behave incongruently, leading to crystallization of Nefiracetam monohydrate in aqueous solution. Ideally, one sees a apparent solubility/dissolution rate of the cocrystal, but the transition occured too rapidly for us to observe. As this is fundamental work, our purpose was to underline cocrystals have the potential to show strong variation in solubility, hence the 2 solvents chosen.

We decided to explain this in the article also.

Line 346

“As the cocrystals considered here behave incongruently in water, with Nefiracetam monohydrate crystallizing out rapidly, we were unable to determine the apparent cocrystal solubility in this solvent. We therefore decided to perform the dissolution studies in EtOH and MeCN as we aim at underlining the potential impact cocrystals can have on physicochemical parameters.”

  1. ZnCl2 choice

Since ZnCl2 belongs to the FDA GRAS list (https://www.accessdata.fda.gov/scripts/fdcc/?set=SCOGS&sort=Sortsubstance&order=ASC&startrow=1&type=basic&search=zinc) and then is pharmaceutically acceptable (see line 223) the use of such a coformer makes sense to us. Besides, literature reveals that racetams are well designed (the γ-lactam) to form an ICC with inorganic salt as ZnCl2 (line 42).

Here is one of the paper related to ZnCl2 and levetiracetam if you may be interested in such paper.

https://doi.org/10.1039/C8CC06199H

  1. References

Are you referring to the “Error! Reference Source not Found” ?

If you do, this matter was already fixed. I am deeply sorry for it.

Again, I am grateful for your good reviewing 

Have a nice evening and stay safe

Xavier Buol

Reviewer 4 Report

Manuscript illustrating extensive screening of nefiracetam cocrystallisations with several of coformers and physicochemical characterisations for 3 cocrystals have been reported. Interesting findings are that a large number of coformers have been screened for cocrystal formation of a drug, which has not been used earlier for any cocrystallisation studies. The physicochemical trends shows some improvements with pharmaceutically acceptable coformers, citric acid, oxalic acid and ZnCl2. However, the manuscript presentation is misleading that it has reported 13 cocrystals based on their basic physical characterisation and no further studies of 10 other cocrystals were presented in this contribution. In my opinion, authors should seriously revise the manuscript to reflect the actual study. Hence, I would recommend a major revision before it is considered for publication.

Comments:

When a reader reads an abstract that for the first time 13 new cocrystals are reported in this work, they would expect to see the physicochemical characterisations and unequivocal identification of those cocrystals. In this manuscript either authors need to consider all 13 cocrystals physical characterisations including if they have single crystal X-ray diffraction studies for them and discuss in detail the solubility and dissolution behavior for 3 cocrystals, which they did already. 

Alternatively, authors need to explicitly say that they have screened 133 coformers, of which 17 or 13 indicated some cocrystal possibility, however, in this contribution 3 cocrystals details study has been presented.

Authors could mention the stoichiometry of those 10 other cocrystals, which shows they have the structural characterisation data as well. It is upto authors to decide if they would consider as part of this work or not. Accordingly, Figure 2 and Table 1 and the manuscript presentation will need to modify.

Either 3 cocrystals study only presented in the manuscript with background screening overview and resubmitted so it will be evaluated if it is suitable for a publication or not. Alternatively, they should consider 13 cocrystals detailed characterisation and re-submitted a manuscript. 

Author Response

Dear Reviewer, 

We thank the reviewer for his suggestion, but do not believe it to be a good idea to remove the screened compounds. This is often missing in scientific papers and asked for by reviewers. We also do not feel like adding the other compounds back into the original paper as this will bring us in direct opposition of the other reviewers who agreed with the choice we did, as readibility will be strongly impacted.

We therefore prefer keeping our initial choice of strategy as is.

Have a good evening and stay safe

Tom Leyssens and Xavier Buol

Reviewer 5 Report

The group of Leyssens and Buol have a presented an exemplary study on the relevance of pharmaceutical co-crystals. Although the improvements in solubility where minor for some of the co-formers, the dissolution rate was notably improved upon. The multi-pronged analysis and synthesis is what is expected nowadays from co-crystal screening for pharmaceutically relevant compounds and this report is a model of it. I recommend acceptance as is.

Author Response

Dear Reviewer

I am so grateful to you for the nice feedback you gave us. I am very glad you appreciate the manuscript.

I will take care of the spelling mistakes as you required.

Thank you for the reviewing and I wish you a nice day

Stay safe

Xavier Buol

Round 2

Reviewer 4 Report

While manuscript work is detailed and interesting, authors seem not attended the comments of one of the reviewers who has asked revision of results presentation.

In the manuscript, thermal, PXRD have been used for an identification of new cocrystal forms/ suspected cocrystals, however, it is difficult to predict stoichiometry from PXRD and DSC. New DSC trace can be due to a polymorph of API/ coformer or it could be a salt, however, authors could able to confirm from their screening studies that what they have got is 1:1 or 2:1 or 4:1:1 and cocrystal polymorphs. It is difficult to understand how they were able to get to that precision of information from their PXRD and DSC - it is certainly incomplete information or inconclusive evidence for their cocrystal formations.

It was strange to see that authors did not mention 2:1, 1:2, and multiple other combinations (3:1, 1:3, 1:4, 4:1), however, they have obtained varying stoichiometries in their screening itself? It certainly looks something missing. Whether the authors carried out solution crystallisations and determined SCXRDs for those 13 cocrystals but they are not disclosing in this work? It is very confusing to reader to understand this set of results.

If authors have 133 coformers screening data, do you consider including either PXRD or DSC data in the supplementary information? Authors should explain how they selectively pick and choose different stoichiometries with a selectively few coformers (2:1, 4:1:1 and so on but they did not try other stochiometries - 1:2, 3:1, 1:3, 1:4)? 

It appears that authors did not understand fully the comments of one of the reviewer who has asked for a major revision. The reviewer did not ask for not to include screening studies, which is a strength of this contributions, however, logical flow of arriving at 13 cocrystals confirmed with definitive cocrystal formation that include (racemic formation) looks very odd to claim from their simple thermal and PXRD data sets.

In the abstract they should expand the fact that they have screened API with 133 coformers, of which 17 coformers show some distinctively different traces/ peaks, which could be potentially cocrystals. Of which 3 cocrystals were fully studied as they already stated can continue.

With the information they revealed it is simply impossible to say to the precision they presented about 13 cocrystals. This should be addressed before it is published. Two other referees coincidentally did not spot this discrepency does not mean, what authors give inconlcusive evidences for the claims are correct. In fact, I agree with a reviewer, who rightly proposed the readability/ results to experimental evidences provided is insufficient at this time. 

Author Response

The authors needed to take the comments of 5 reviewers into account, 4 of which agreed with the way the results were presented. This is the reason the authors decided not to pursue changing the way the results were presented.

  • As stated in the text, the stoichiometry of the cocrystals is fully determined by single crystal XR as commonly done in litterature. The single crystal files are added and undoubtly give access to the stoichiometry. There is no claim in the paper stating that the stoichiometry is determined based on DSC and XRPD.
  • We added a clarification to the text 'Grinding a 1:1 stoichiometric amount of parent compounds together, a observation of new peaks is indicative of cocrystal formation, but could potentially be due to phase transitions (eg. polymorphic transition).  .... To determine cocrystal formation as well as the exact stoichiometry of the cocrystal, single crystals were grown'.  We hope this satisfies; Cocrystal screening is typically done only using a 1:1 stoichiometry. In case a 1:2 or 2:1 cocrystal exists, novel peaks will appear together with the peaks of the excess parent compound. The XRPD screening is merely used as an indicaiton of possible cocrystal formation, as now specified in the text.
  • It is the authors believe that this reviewer (contrary to the other 4 reviewers did not grasp the text correctly). As the other reviewers clearly understood, we have 'SINGLE CRYSTAL' proof of 13 cocrystals and the CIF files were even added to the original submission. So there is no doubt on the existence of these cocrystals. THe reviewer likely did not realize this.